# Roles of NAD^+^ in Acute and Chronic Kidney Diseases

**DOI:** 10.3390/ijms24010137

**Published:** 2022-12-21

**Authors:** Marya Morevati, Evandro Fei Fang, Maria L. Mace, Mehmet Kanbay, Eva Gravesen, Anders Nordholm, Søren Egstrand, Mads Hornum

**Affiliations:** 1Department of Nephrology, Rigshospitalet, University of Copenhagen, 2100 Copenhagen, Denmark; 2Department of Clinical Molecular Biology, University of Oslo and Akershus University Hospital, 1478 Lørenskog, Norway; 3Division of Nephrology, Department of Medicine, Koç University School of Medicine, Istanbul 34010, Turkey; 4Department of Pathology, Herlev Hospital, University of Copenhagen, 2730 Copenhagen, Denmark

**Keywords:** NAD^+^, mitochondria, autophagy, acute kidney injury, Klotho, chronic kidney disease, fibrosis, senescent

## Abstract

Nicotinamide adenine dinucleotide (oxidized form, NAD^+^) is a critical coenzyme, with functions ranging from redox reactions and energy metabolism in mitochondrial respiration and oxidative phosphorylation to being a central player in multiple cellular signaling pathways, organ resilience, health, and longevity. Many of its cellular functions are executed via serving as a co-substrate for sirtuins (SIRTs), poly (ADP-ribose) polymerases (PARPs), and CD38. Kidney damage and diseases are common in the general population, especially in elderly persons and diabetic patients. While NAD^+^ is reduced in acute kidney injury (AKI) and chronic kidney disease (CKD), mounting evidence indicates that NAD^+^ augmentation is beneficial to AKI, although conflicting results exist for cases of CKD. Here, we review recent progress in the field of NAD^+^, mainly focusing on compromised NAD^+^ levels in AKI and its effect on essential cellular pathways, such as mitochondrial dysfunction, compromised autophagy, and low expression of the aging biomarker αKlotho (Klotho) in the kidney. We also review the compromised NAD^+^ levels in renal fibrosis and senescence cells in the case of CKD. As there is an urgent need for more effective treatments for patients with injured kidneys, further studies on NAD^+^ in relation to AKI/CKD may shed light on novel therapeutics.

## 1. Introduction

Nicotinamide adenine dinucleotide (the oxidized form NAD^+^) was first discovered in 1906 as a cellular component enhancing alcohol fermentation [1]. In recent years, NAD^+^ has been shown to have multiple functions, such as exhibiting an essential role in mitochondrial redox reactions and operating as an essential cofactor for PARylation (addition of poly-ADP-ribose to proteins), cyclic-ADP-ribose (cADPR), and sirtuin (Sirt)-dependent deacetylation [2,3,4]. NAD^+^ is a metabolite found in limited amounts in cells [5,6,7,8,9], and its level decreases with age. NAD^+^ homeostasis is linked to extended lifespan and enhanced resistance to infectious and inflammatory diseases [5,6]. In addition, it is thought to prevent the development of diseases such as kidney disease, cardiovascular and neurodegenerative disease, diabetes, and metabolic syndrome [10,11,12,13,14,15,16,17].

Acute kidney injury (AKI) causes approximately two million deaths each year worldwide [18,19] among patients with generalized or localized renal ischemic injury due to surgery, sepsis or trauma, and toxic responses to medication (often in combination) [20,21,22]. AKI incidence increases with age [23,24,25]. Partial recovery of AKI or repeated incidents of AKI are independent risk factors for the development of chronic kidney disease (CKD) [26]. A range of studies has reported a decline in the level of NAD^+^ in AKI and CKD cases [27,28,29]. One approach to increasing NAD^+^ levels in rodents and humans is thought to be supplementation with NAD^+^ precursors, such as nicotinamide (NAM), nicotinic acid (NA), tryptophan, nicotinamide mononucleotide (NMN), or nicotinamide riboside (NR). Supplementation with NAD^+^ improves cellular NAD^+^ homeostasis, and studies suggest that it may potentially delay the aging process [5,30,31]. With this background in mind, we review recent progress in the field of NAD^+^, focusing on: AKI and aging-related biomarkers, including mitochondrial dysfunction, impaired autophagy, reduced level of the kidney biomarker Klotho, and occurrence of renal fibrosis and senescence cells; the progression to CKD; and NAD^+^ as a potential pharmacological option. 

## 2. The Role of NAD^+^ and NADH Redox Cycling in Mitochondria

NAD molecules exist in an oxidized (NAD^+^) and reduced (NADH) form, known as a redox couple. NAD^+^ is an electron carrier for oxidative reactions in the nuclear, cytosolic, and mitochondrial compartments [32,33]. This metabolite is necessary for the constant flow of electrons required for intermediary metabolic functions such as the oxidation of fuel substrates by glycolysis, the tricarboxylic acid (TCA) cycle, and the β-oxidation of fatty acids. In glycolysis, the oxidation of glyceraldehyde-3-phosphate to 1,3-bisphosphoglycerate is linked to the reduction of cytosolic NAD^+^ to NADH. During aerobic respiration, the malate–aspartate shuttle transfers electrons into mitochondria. Mitochondrial malate dehydrogenase catalyzes the donation of the electrons to the NAD^+^ to form the NADH and oxaloacetate, which then enters the citric acid cycle via transamination. The transamination converts it to aspartate, which can then leave the cell via the anti-transporter that exchanges glutamate for the aspartate (which donates an amino group to oxaloacetate to create acetate in the transamination step). Once the glutamate has donated the amine, it forms alpha-ketoglutarate, which is then used to exchange malate via the antiporter system and is further transported to the mitochondrial electron transport chain (ETC), which generates cytosolic NAD^+^.

## 3. NAD^+^ Homeostasis

NAD^+^ has a half-life of 1–2 h in the cytoplasm and nucleus and about 8 h in mitochondria [34,35]. Even still, the intracellular concentration of NAD^+^ appears to be adequately stable in healthy states with only a few-fold change during metabolic activity [36]. The NAD^+^ level in the cell is in constant flux because of the many pathways involved in NAD^+^ biosynthesis and consumption (Figure 1A,B). Three biosynthetic pathways regulate the cellular NAD^+^ level: (1) the de novo synthesis pathway, (2) the Preiss–Handler pathway, and (3) the salvage pathway [15]. 

## 4. NAD^+^ Biosynthesis

### 4.1. De Novo Biosynthesis

The de novo biosynthesis pathway converts dietary tryptophan to NAD^+^. The synthesis starts with tryptophan 2,3-dioxygenase (TDO) or indoleamine 2,3-dioxygenase (IDO). The two tryptophan-degrading enzymes initiate the formation of the unstable intermediate product α-amino-β-carboxymuconate-ε-semialdehyde (ACMS), which can be either spontaneously cyclized to quinolinic acid (QA) or be converted by the decarboxylase (ACMSD) into α-aminomuconate-ε-semialdehyde (AMS), which can then be converted into picolinic acid. Katsyuba et al. showed that ACMSD limits the formation of NAD^+^ via ACMS conversion [37]. The other important step in catalyzing QA to NAMN is via quinolinate phosphoribosyltransferase (QPRT), which carries out the pathway to NAD^+^ biosynthesis via the Preiss–Handler pathway [38,39,40,41]. Mehr et al. showed that QPRT is essential for the homeostasis of NAD^+^ in the kidney, which implies an important function for NAD^+^ synthesis [41]. Furthermore, it has been reported that endoplasmic reticulum (ER) stress impairs de novo NAD^+^ biosynthesis, as the transcription factor DNA damage-inducible transcript 3 (DDIT3) may suppress QPRT function [40]. De novo NAD^+^ biosynthesis was observed to be impaired in many studies in AKI animal models and humans [41,42].

### 4.2. Preiss–Handler Pathway 

The Preiss–Handler pathway can convert dietary nicotinic acid (NA) to nicotinic acid mononucleotide (NAMN) by using the enzyme nicotinic acid phosphoribosyltransferase (NAPRT) [43]. NMNAT further converts NAMN to nicotinic acid adenine dinucleotide *(*NAAD), and NAD synthase (NADSYN) subsequently converts NAAD to NAD^+^ by using glutamine as a nitrogen donor [44,45]. 

### 4.3. Salvage Pathway

The salvage pathway is the main pathway used in NAD^+^ synthesis, retrieving NAD^+^ from the recycling of NAM, NA, NR, and NMN [46]. NAM created by NAD^+^ consumption pathways is recycled into NMN by nicotinamide phosphoribosyl transferase (NAMPT), which is the rate-limiting reaction in the salvage pathway [47]. The NAMPT level is dynamic and responds to the cellular demands for NAD^+^ and cell stresses such as DNA damage and fasting [48]. The NAD^+^ precursor, NR, is imported by equilibrative nucleoside transporters (ENTs) and transformed to NMN by NR kinase 1/2 (NRK1,2). Ultimately, NMN is adenylated by nicotinamide mononucleotide adenylyl transferases (NMNAT) to yield NAD^+^ [49,50]. NMNAT 1 and 3 have been shown to be significantly downregulated in CKD rat models, and thereby this pathway may be impaired in CKD [51]. 

## 5. Consumption of NAD^+^

### 5.1. PARP1 

PARPs belong to a family known as APD-ribosyl transferases [52]. They are known to have several roles in the cell including nuclear DNA metabolism, as well as being implicated in mRNA splicing, cellular response to stress, and regulation of circadian rhythm [53]. PARP1 is the most investigated protein in this family and is often called the guardian of the genome [52,54]. Cellular stress and especially DNA lesions trigger the DNA damage sensor protein, PARP1, which consumes NAD^+^ for loosening up chromatin at specific loci to enhance DNA repair and for PARylation, a process by which PAR chains are generated to serve as a docking platform for the recruitment of other DNA repair proteins, with high biochemical complexity. Recently, Lee et al. (2022) provided evidence for the existence of PARP1 in mitochondria and showed that PARP1 directly binds to mitochondrial DNA (mtDNA) and causes NAD^+^-dependent mitochondrial nucleoid PARylation, which stimulates mtDNA transcription [55]. 

Furthermore, activation of PARP1 can also enhance cell death to maintain genomic integrity [56,57,58]. However, severe or persistent DNA damage in model organisms lead to hyperactivation of PARP1 and, consequently, the depletion of the limited cellular metabolite NAD^+^ [59]. 

### 5.2. cADPR Synthetases

cADPR is a nucleotide found in many cell types across species. cADPR is known as one of the three main second messengers for intracellular Ca^2+^ mobilization together with inositol 1,4,5-trisphosphate and nicotinic acid adenine dinucleotide phosphate (NAADP) [60]. cADPR-induced intracellular Ca^2+^ mobilization is involved in diverse cell functions, including cell proliferation, differentiation, fertilization, muscle contraction, and secretion of neurotransmitters, hormones, and enzymes [60]. CD38 and CD157 are enzymes that utilize NAD^+^ to generate secondary messenger molecules, including ADPR, 2-deoxy-ADPR (2dADPR), NAADP, and cADPR [61,62]. CD38 expression increases with age, which is assumed to be associated with the reduced level of NAD^+^ in aging [61,63]. 

Sterile alpha and TIR motif containing 1 (SARM1) is another well-known enzyme in this pathway that utilizes NAD^+^ and catalyzes NAM, ADPR, and cADPR in neurons [64]; SARM1 is also highly expressed in the kidney. However, its NADase activity was not investigated [65]. 

### 5.3. Sirtuins

The Sirtuins (SIRTs) are a family of NAD^+^-dependent deacetylase and mono-ADP-ribosyltransferases that are likely to be found in the nucleus (Sirt6, and Sirt7), mitochondria (Sirt3-5), and nucleus and cytoplasm (Sirt1 and Sirt2), respectively [66,67,68,69]. The seven Sirt homologs characterized in mammals have different substrate preferences, enzymatic activities, and targets [66,67]. All SIRTs share a common conserved NAD^+^-binding domain but differ in their amino and carboxy-terminal regions, catalytic activity, and cellular functions [66,67]. They impact inflammation, cell growth, circadian rhythms, energy metabolism, and stress resistance [70,71,72,73,74,75,76,77,78]. Sirt1 is the most investigated homolog of this family and is primarily located in the nucleus [79]. Activation of Sirt1 results in the deacetylation of downstream target proteins, such as DNA repair proteins, including Ku70 [80], PARP1 [81], and Werner syndrome ATP-dependent helicase (WRN) [82]. Sirt1 can also indirectly regulate mitochondrial biogenesis and antioxidants such as peroxisome proliferator-activated receptor-gamma coactivator-1 alpha (PGC1-α) and forkhead box O (*FOXO*) transcription factors [83,84,85,86]. Loss of renal tubular PGC1-α enhances susceptibility to AKI and long-term renal fibrosis [87,88]. An experimental study showed that activation of PARP-1, a high consumer of NAD^+^, limited the availability of NAD^+^ for Sirt1 activity [89], indicating that PARP-1 and Sirt1 may play equivalent but inverted roles in the regulation of cellular energy use. Furthermore, Sirt1 and Sirt3 are also shown to have reno-protective effects through several pathways by improving mitochondrial function, reducing oxidative stress, and inhibiting transforming growth factor-beta (TGF-beta) signaling [90,91,92,93]. Sirt5 has also been shown to improve renal function in cisplatin-induced AKI by regulation of nuclear factor-erythroid factor 2-related factor 2/heme oxygenase*-*1 (Nrf2/HO-1) and B-cell lymphoma 2 (Bcl-2) in human kidney HK-2 cells [94]. In addition, Sirt7 deficiency has been shown to reduce inflammation and tubular damage induced by bilateral ischemia-reperfusion injury (IRI) for 22.5 min compared to two-month-old control mice [95] and ameliorates cisplatin-induced AKI through regulation of the inflammatory response [96]. Another study in mice showed that administering histone deacetylase inhibitors could reduce fibrotic and inflammatory processes [97]. 

### 5.4. Nicotinamide N-methyltransferase (NNMT)

NNMT is abundantly expressed in the kidney [98,99] and is implicated in the progress of age-associated diseases such as cancer and obesity through the consumption of methyl donors, decreased NAD^+^ content, and increased active NAD^+^ metabolites [100,101]. NNMT catalyzes the transfer of a methyl group from S-adenosylmethionine (SAM) to NAM, which results in S-adenosylhomocysteine (SAH) and methyl nicotinamide (MNA) [100,101]. Subsequently, MNA is metabolized to N-methyl-2-pyridone-5-carboxamide (N-Me-2PY) and N-methyl-4-pyridone-3-carboxamide (N-Me-4PY) [102]. N-Me-2PY is known to be a uremic toxin with repressive potency toward PARP and accumulates in CKD [103,104]. N-Me-2PY was also shown to become elevated in human plasma with age [105]. Therefore, supplementation with NAM in CKD patients may increase kidney toxicity via the accumulation of N-Me-2PY. NAM administration for 24 weeks was shown to elevate the level of N-Me-2PY more than five times in the CKD group compared with the non-treated control [104]. However, an additional group with a placebo was missing in this study. Another study found that N-Me-2PY and NAM inhibited PARP1 activity in vitro, with IC50 values of 8 µM and 80 µM, respectively [104]. Those studies indicate that short-term exposure to elevated N-Me-2PY levels may exert protective effects, whereas prolonged exposure is probably harmful due to the importance of DNA repair [105]. Furthermore, NAM catabolites may inhibit several crucial enzymes for cell division and proliferation. 

## 6. Kidney Function and NAD^+^

The kidney is one of the organs with the highest level of cellular NAD^+^ [106]. Cellular control of NAD^+^ is crucial for renal metabolic and bioenergetic homeostasis [107,108]. The renal tubule cells require a high energy level for the reabsorption of the filtrate obtained by establishing energy-intensive electrochemical gradients over the apical and basolateral membrane between the filtrate and the vasculature [109,110]. Mitochondria are the powerhouses of the cell and are essential for homeostatic control of renal function. A significant amount of ATP in the renal cortex is generated by mitochondrial consumption of NADH, which is formed by the TCA cycle and β-oxidation [111]. Dysfunction in mitochondrial respiration causes a cytosolic reduction of pyruvate to lactate and converts NAD^+^ to allow for ongoing glycolysis in the kidney. 

Furthermore, low levels of NAD^+^ and dysfunction of the mitochondrial ETC are the leading cause of ROS production [112]. ROS at low levels regulates several cellular processes such as cell proliferation, survival, and growth; however, high levels of ROS causes activation of pathways, including P53/P21 signaling and mitogen-activated protein kinase pathways, which leads to the elevation of apoptosis, inflammation, and stress-induced senescence [113,114]. In addition, it has been demonstrated that ROS cause telomere-associated genomic instability and senescence in mesenchymal stem cells [115,116]. 

NAD^+^ is also shown to be essential for kidney nephrogenesis and function, as identified mutations in NAD^+^ de novo biosynthesis pathway genes, namely *Kynureninase* (*KYNU*) and *3-hydroxyanthranilate 3,4-Dioxygenase* (*HAAO*), promoted multiple congenital malformations, causing dysplastic kidney in patients [117]. Furthermore, an elevated urinary quinolinate-to-tryptophan (Q/T) ratio in post-cardiac surgery patients has shown to be a good indicator of a reduced QPRT activity, compromised de novo NAD^+^ biosynthesis pathway in the kidney, and a useful early marker of AKI [40,41].

A range of studies has shown that NAD^+^ levels are significantly reduced in the case of AKI (Table 1) [41,107,118]. However, as NAD^+^ is involved in many redox reactions and functions as a cofactor for many enzymatic reactions, it would not be appropriate to specify a single downstream effector pathway in AKI. Instead, AKI can be evaluated as a temporary state of reduced NAD^+^ homeostasis with several damaging results for injured cells.

## 7. Acute Kidney Disease and NAD^+^ Level

AKI is defined as the immediate disruption of function and damage to the tissue structure of the kidney. Multiple factors cause AKI, and many patients with AKI have a combined etiology where the presence of sepsis, ischemia, and nephrotoxicity frequently co-exist. Several biomarkers have been proposed to diagnose AKI at different levels of progression and verification [120,121,122,123]. For instance, clinical biomarkers such as creatinine and cystatin C, the most studied ones for tubular damage such as NGAL and KIM-1 and suPAR for damages in podocytes; for experimentally usage beta trace, B2-microglobulin, Klotho, and much more are used. However, it is unclear if a single or multiple biomarker approach is sufficient to diagnose the complicated and multifactorial aspects of AKI [124,125,126,127].

The primary diagnostic approach in hospitals today for AKI rests on an acute decrease in the glomerular filtration rate (GFR), as reviewed by an acute rise in serum creatinine levels and/or a decrease in urine output over a given time interval [128,129]. 

AKI can be induced in animal models by IRI, which mimics the clinical situation of hypotension-induced AKI. Our group and others have shown that the level of NAD^+^ was decreased in the IRI-induced AKI animal model compared to controls (Figure 2) [118,119]. Furthermore, the IRI animal model had compromised autophagy, mitochondrial function, and decreased expression of Klotho [118,130]. 

In addition, another study has shown that a significantly low level of NAD^+^ was observed in the kidneys of 20-month-old mice compared to 3-month-old mice with AKI [119]. Those observations demonstrate that the regulation of NAD^+^ fundamentally impacts AKI and healthy aging. 

### 7.1. Compromised NAD^+^ Homeostasis in AKI 

We recently showed that IRI, a model of AKI, causes a decreased level of NAD^+^ in the kidney [118]. In this model, there was also a low expression of *Sirt1*, *Klotho*, and the antioxidants *Sod2* and *catalase* [118] (Figure 2). These findings may indicate that low NAD^+^ and *Sirt1* expression results in impaired mitochondria function, which probably leads to an elevated level of ROS and a low level of antioxidants in the cell. The impaired mitochondrial function is shown to be rescued by Sirt1 activators such as SRT1720 [131]. The Sirt1 activator leads to deacetylation and activates PGC1-α, which results in a recovery of mitochondrial protein expression and function, and normal mitochondrial biogenesis [132,133]. In addition, overexpression of PGC1-α in renal tubular cells was shown to elevate the NAD^+^ through the de novo pathway. NAD^+^ can also post-translationally activate PGC1-α [134], which indicates that PGC1-α and NAD^+^ might reciprocally regulate each other.

### 7.2. Disrupted FAO and Level of NAD^+^ in AKI 

Several studies have shown that a low level of NAD^+^ and compromised mitochondrial function leads to briefly disrupted fatty acid oxidation (FAO) during AKI [107,135,136,137]. FAO dysfunction is known to be a major driver of renal fibrosis through pathways such as STAT6 contribution [138]. Therefore, properly recovering the FAO after insult is crucial, as a defect in renal FAO leads to lipid accumulation and/or reduction of FAO likely participates in the mesenchymal reprogramming of renal epithelial cells, elevating the risk of developing CKD from even transient AKI [139,140,141]. This phenomenon indicates that NAD^+^ plays a significant role in the progression of renal fibrosis through the regulation of FAO. The defect in FAO and increased desaturation of polyunsaturated fatty acids (PUFAs) to highly unsaturated fatty acids (HUFAs) because of NAD^+^ insufficiency and compromised mitochondrial function result in the buildup of HUFA-accommodated triglycerides and cellular lipids in renal tubular cells. In addition, PGC1-α overexpression in mice elevates FAO levels [107].

### 7.3. Induced PARP1 in AKI and the Level of NAD^+^

Pathologically induced PARP1 in AKI is associated with renal cell stress and accelerated consumption of NAD^+^ [142,143]. A study in rats subjected to IRI showed an increase in the level of PARP1 expression in the renal tubules within 12 h post-reperfusion, and administration of a PARP inhibitor after reperfusion caused an elevated level of ATP, diminished histopathological indication of tubular damage, and improved renal function in this model [142]. In line with this finding, PARP1 knockout mice were more resistant to renal IRI than the control animals [144]. These results suggest that PARP activation is a part of the cascade of molecular events that occurs after IRI in the kidney. Whether suppression by PARP inhibitors (PARPi) olaparib, rucaparib, and talazoparib are beneficial in clinical AKI requires further study [145,146]. PARPi is suggested to function by two mechanisms: catalytic inhibition of PARP or “PARP trapping” (PARP is trapped at sites of DNA damage), which results in the elimination of repair and cytotoxicity [147]. In vitro, it has been shown that olaparib and talazoparib are more effective at lowering cell survival, proliferation, clonogenic capacity, and xenograft growth compared with those inhibitors (veliparib) that selectively inhibit PARP catalytic activity [148]. However, the use of PARPis must be proceeded with caution, as PARP has an essential role in DNA repair, which can be important in the recovery phase of AKI.

### 7.4. Elevated Expression of CD38 in AKI

In our recently published study on IRI-induced AKI rats, we also observed a significant increase in the level of CD38 after 24 h and 14 days post-IRI, which we suggested to be a factor that reduces the level of NAD^+^ in post-IRI subjects [118]. In addition, other studies have shown that inhibiting CD38 increases the NAD^+^ level in the tissue and promotes metabolic changes, leading to increased longevity in aged mice [61,149]. Furthermore, CD38 inhibition also causes a reduction of lipopolysaccharide-induced macrophage M1 polarization and AKI, which suppress the activation of the proinflammatory signaling by nuclear factor-kappa B (NF-κB) [150]. Therefore, targeting CD38 may provide an efficient strategy for retarding the initiation and progression of AKI. Inhibition of CD38 by a distinct CD38 inhibitor, 78c, has been shown to elevate protein deacetylation [149]. Furthermore, genetic elimination or inhibition of CD38 has been shown to fortify mice from conditions that damage Sirt1 activity, such as high-fat diets [151], dysregulation of glucose and lipid homeostasis in obesity [152], age-associated mitochondrial dysfunction [61], and d-galactose-induced myocardial cell senescence [153]. Therefore, the interplay between CD38 and SIRTs is an important component of the pathophysiology of diseases associated with NAD^+^ decline (Figure 3).

### 7.5. Decreased Level of Sirt1 in AKI 

NAD^+^ depletion also caused the inhibition of Sirt1 and its downstream target genes, which mediates mitochondrial quality control and renal tubular ATP production level. *Sirt1* is highly expressed in the kidney, especially in the renal medulla, and Sirt1 deficiency elevates AKI susceptibility in mice [119]. A high level of Sirt1 is shown to ameliorate albuminuria in diabetic patients, lower blood pressure, prevent cardiovascular-related AKI, delay kidney fibrogenesis, develop cyst generation, and prevent renal aging [154,155,156,157]. In a study with a cisplatin-induced AKI model, the role of Sirt1 was examined in a novel transgenic mice model with a proximal tubules-specific overexpression of *Sirt1*. *Sirt1* overexpression in mice was shown to maintain the number of peroxisomes and their catalase activity, which was highly decreased in the control groups [158]. Based on this, Sirt1 may have a protective effect on tubular cell damage and kidney injury [159]. Elevated expression of *Sirt1* also promoted cellular recovery in IRI compared with unmodified *Sirt1* in the IRI control [160]. Fan et al. showed that Sirt1 heterozygotes were more vulnerable to IRI and had an elevated level of the cyclin-dependent kinase (CDK) inhibitor, p21 [160]. p21 is a downstream target for p53, which binds and causes inhibition of CDK2 or CDK4 complexes activity and subsequent regulation of cell cycle progression at the G_1_ phase [160]. However, Sirt1 reduces p53, which might suppress p21 and result in cell proliferation.

These studies and an extended range of other studies indicate that the reduced NAD^+^ in renal tissue is probably a combination of accelerated NAD^+^ consumption and reduced NAD^+^ biosynthesis in AKI [12,41,108,161,162,163]. In addition, numerous studies in model organisms, ranging from *Caenorhabditis elegans* to mice and humans, have demonstrated the beneficial effects of NAD^+^ boosting in various diseases [164]. Furthermore, administration with precursors such as NAM ameliorated the mild ischemic AKI in animal models and AKI patients [41,88,119]. However, the mechanisms by which NAD^+^ supplementation is efficacious remain incompletely understood. 

### 7.6. Low NAD^+^ and Compromised Autophagy in AKI 

Reduced levels of NAD^+^ are also associated with compromised autophagy in the kidney. Autophagy is a catabolic process for removing damaged or useless cellular components to maintain cellular homeostasis and is important for health and disease [165]. The first step in autophagy is the formation of the phagophore, a double lipid membrane that engulfs target proteins and organelles from the cytoplasm and forms autophagosomes. The autophagosome subsequently fuses with the lysosome for the degradation of its content, which is known as an autolysosome. 

LC3B and P62 are two of the most investigated biomarkers for autophagy, and their amounts are known to reflect the autophagic activity. In the case of cellular stress, the LC3B-II, which is localized on the phagocytic membrane, becomes degraded in the lysosomes. The level of LC3B-II depends on both autophagosome production and degradation rate. Therefore, an increased level of LC3B-II indicates a high level of autophagy. However, P62 is a substrate for autophagy degradation, and a high level of P62 indicates a low level of autophagy [166]. Our group showed that P62 is significantly increased in the 24 h post-IRI rat model compared to the control. However, fourteen days of prophylactic administration of NR reduced the level of P62 in the 24 h post-IRI rats [118]. Lynch et al. also showed in an in vitro and cisplatin nephrotoxicity mouse model that NMN administration protected against autophagy and enhanced lysosomal disposal of injured mitochondria [167]. These findings indicate that autophagy may have been compromised in AKI and that boosting with NAD^+^ precursor ameliorates the autophagy in those models. Elevated autophagy protects the kidney from toxins and IRI-induced AKI [130,168,169]. Other key proteins in the autophagy pathway regarding the kidney have also been investigated. Beclin1 has been shown to be essential in initiating autophagosome formation [170,171]. A mutation in the BH3 domain of Beclin1 reduces the binding of two negative regulators of autophagy (BCL2 and BCL-XL) to Beclin1 in vitro and in vivo, thereby increasing autophagy [172,173,174,175]. *Klotho* knockout mice, characterized by kidney disease and aging, also exhibited an increase in association with Beclin1 to BCL2, and a knockout of the disrupted Beclin1 improved autophagy and longevity in those mice [174]. These findings indicate that autophagy is essential for longevity and kidney health.

### 7.7. Klotho

Klotho is a well-known biomarker of kidney injury and is mainly derived from the cortex of the kidney. Its expression quickly drops in AKI with reduced levels in urine and blood but is restored after recovery [176]. Klotho has a broad range of effects on renal tubular epithelial cells, such as regulating renal 1,25-(OH)2 vitamin D3 production and phosphate, calcium, and potassium homeostasis [177,178,179]. Deficiency of Klotho leads to increased kidney dysfunction and renal histology damage compared to wild-type rats after IRI; however, overexpression of *Klotho* was shown to protect against renal aging in mice [180,181]. Furthermore, overexpression of *Klotho* was shown to extend the lifespan by 30% in the transgenic mice model compared to the wild-type, and *Klotho*-deficient phenotypes are associated with aging human phenotypes [177,182,183,184]. Exogenous Klotho supplementation also showed increased endogenous Klotho levels and reduced AKI [180,185,186]. 

Shi et al. showed a close network among phosphate, autophagy, and Klotho, which is essential for healthy kidney maintenance. Beclin1 suppresses renal sodium-dependent phosphate transport protein (NaPi-2a/2c) expression, increases phosphaturia, and reduces plasma phosphate. Low plasma phosphate upregulates Klotho, which further combats aging. High plasma phosphate suppresses, while Klotho stimulates autophagy. Autophagy and Klotho interactively counteract phosphotoxicity, prolong lifespan, and attenuate aging-associated multiple organ degeneration [187]. In addition, other studies have shown that resveratrol, a well-known NAD^+^ precursor, elevates the level of *Klotho* gene expression by activating the transcription factor 3/c-Jun complex-mediated signaling pathway in mice [188]. These findings indicate that *Klotho* expression is regulated by Sirt1 and autophagy activity, which are two NAD^+^ dependent pathways. In our study, we showed a decrease in the level of Klotho in the IRI-induced AKI rats; however, we could not see a change in the level of Klotho after treatment with NR [118]. 

## 8. Chronic Kidney Disease

Patients with AKI may gain normal kidney function but have an increased risk of developing progressive CKD [189,190,191]. CKD is characterized by a persistent loss of renal function and affects approximately 10–15% of the world’s population. CKD is a significant global public health issue [192,193,194]. Regardless of the etiology resulting in CKD, the common terminal pathway is renal fibrosis, which is histologically characterized by glomerulosclerosis, tubular atrophy, and interstitial fibrosis [195]. Specific treatments for slowing down CKD progression and preventing CKD-related complications are limited. The progressive nature of CKD may progress to end-stage renal disease (ESRD), which requires renal transplantation or treatment with dialysis [51,196,197].

It has been reported that the impaired de novo pathway, including the altered tryptophan metabolism, is involved in decreased NAD^+^ levels in CKD, and the uremic toxin indoxyl sulfate was shown to reduce the contents of Sirt1 and Sirt3 in the CKD rat model [198,199,200]. Kumakura et al. (2021) showed that NAM treatment lowers the risk of developing renal failure in an adenine-induced CKD mouse model [29].

In a study with 205 CKD patients with an eGFR of 20–45 mL/min/1.73 m^2^, the intake effect of NAM, lanthanum carbonate, or NAM in combination with lanthanum carbonate or placebo for 12 months was investigated. However, the result showed no significant change in those patient’s mineral metabolism parameters or the mean of estimated GFR from baseline at the 12-month follow-up between the study groups [28]. Because of possible severe damage to kidney cells, these findings suggested that boosting NAD^+^ by NAM may not be beneficial in CKD cases. Furthermore, a study in CKD rat models, a 5/6 nephrectomy (5/6 Nx) model and an adenine model, respectively, showed that the level of renal NAD^+^ and NADH as well as the expression of the three key enzymes in NAD^+^ biosynthesis, QPRT, and NMNAT1/3 was significantly downregulated in the kidney of the two CKD models. In addition, Sirt3 and CD38 were downregulated in the CKD rat models [29]. NAM and NR are converted to NMN and then to NAD^+^, which is synthesized under the catalysis of NMNATs. This study shows that NMNAT 1/3 are both downregulated in CKD models. Therefore, this study suggests that augmentation with NMN or increasing NMNAT 1/3 expression may be beneficial in delaying the progression of CKD [51]. However, the current conclusion about NAD^+^ boosting and CKD is inconclusive.

### 8.1. Fibrosis in CKD

Different types and severities of kidney injuries are shown to have different effects on cell cycle arrest [201]. In particular, Yang et al. showed that the induction of moderate IRI disrupts renal function, although the kidney function returns to normal after seven days. However, severe IRI and nephrotoxic insults (acute aristolochic acid toxic nephropathy) result in a higher degree of damage in the kidney, which causes delayed recovery and leads to long-term G2/M-arrested cells and subsequently to significant interstitial fibrosis in the kidney [201]. Tubulointerstitial fibrosis is the most common feature of progressive kidney disease and is caused by an accumulation of matrix proteins such as collagen I, which is the most expressed matrix protein in renal fibrosis. Other studies revealed that adhesive glycoprotein fibronectin (FN) is also found [202,203,204]. Our group recently showed that circulating endotrophin, a collagen VI precursor, was involved in the pathogenesis and was a prognostic for mortality after AKI [205]; it was also involved in the development of fibrosis following IgA nephritis and ANCA vasculitis [206].

Renal fibrosis affects all kidney compartments and extends to a range of kidney diseases. Renal fibrosis is a typical feature of ESRD and is one of the principal mechanisms involved in AKI to CKD transition [207,208,209]. Although there is a broad understanding of the involvement of tubular pathologies, inflammation and infiltration of inflammatory cells, activation and expansion of fibroblasts, and compromised microvasculature [207,208], the pathogenesis of renal fibrosis is still not completely elucidated. More studies are required to understand the mechanism underlying the development of renal fibrosis.

In addition, some studies have shown that senescence markers correlate with the amount of kidney fibrosis in mice [210,211,212]. Luo et al. showed that Wnt9a promotes renal fibrosis by accelerating cellular senescence in tubular epithelial cells in rats [213], and persistent injury and chronic inflammation are also shown to induce senescence. Senescent cells may likely be involved in profibrotic circumstances through proinflammatory cytokines, especially IL-6 and IL-8, which are essential regulators in the induction of senescence; their senescence-associated secretory phenotype (SASP) may also be involved in aging and renal diseases. However, they are also required for the activation of inflammatory responses to remove pathogens and mediate tissue repair.

Mitochondrial dysfunction has also been observed in CKD, and preventing mitochondrial damage reduces fibrogenesis [214,215,216,217]. Several treatment options for mitochondrial dysfunction have been suggested [218]. However, the exact role of cellular senescence in renal fibrosis is not well investigated.

### 8.2. NAD^+^ and Fibrosis 

The inflammatory response of renal tubular epithelial cells is key to developing renal interstitial fibrosis [208,219,220]. Treatment with exogenous NAM in murine with unilateral ureteral obstruction (UUO), a model of renal fibrosis, showed that NAM suppresses renal interstitial fibrosis by reducing tubular injury and related inflammation, such as the expression of proinflammatory cytokines TNF-α and IL-1β. In addition, in mice-cultured renal proximal tubule cells, NAM caused aninhibition of TGF-beta1, a master regulator of myofibroblast differentiation in fibrosis. Furthermore, NAM showed a pronounced inhibitory effect on fibrosis proteins such as FN and alpha-smooth muscle actin (α-SMA) expression [27]. This model reveals the extent to which NAD^+^ homeostasis can protect against tubular atrophy and interstitial damage. 

In addition, chronic or persistent inflammation is another essential factor driving the progression of renal fibrosis [208,219,221,222]. Faivre et al. showed that mice with a prophylactic administration of NR for seven days before being subjected to UUO or chronic proteinuria indicated that augmentation with NR did not improve GFR or the level of plasma urea or creatinine despite boosting the kidney NAD^+^ content in chronic proteinuria mice [223]. NR failed to protect interstitial renal fibrosis, as the gene expression of the metabolic, proinflammatory, and profibrotic pathway was unchanged in those models. Further studies are needed to investigate PARP-1 and SIRTs in experimental models of renal fibrosis for a better understanding of its anti-fibrotic effect. Sirt1 is known to deacetylate the P65 subunit of NF-κB and inhibit NF-κB proinflammatory signaling and the production of FN [224]. Decreased acetylated Sirt6 in the proximal tubule decreases renal fibrosis [225]. Furthermore, it has been shown that augmentation with NMN and NAM improves albuminuria in diabetic nephropathy and promotes renal fibrosis in diabetic mice [27,226,227]. Those studies indicate the decreased levels of NAD^+^ in CKD and promote CKD by elevating NAD^+^.

Takahashi et al. also recently showed that NAD^+^ metabolism is changed in human and mouse models of CKD. The final metabolites of NAD^+^, N-Me-2PY, and N-Me-4PY were observed to be elevated in the serum of CKD patients compared to healthy controls. These metabolites are reported to be uremic toxins, which cause renal cell damage [103,228]. Furthermore, NNMT knockout mice that developed renal fibrosis showed a decrease in renal NNMT expression and NAD^+^ salvage pathway metabolites such as NAM, NMN, and NAD^+^, as well as an increase in degradation products of NAM such as MNA, N-Me-2PY, and N-Me-4PY. In addition, the NNMT knockout mice have a decreased level of renal profibrotic genes and decreased level of inflammation. This result indicates that NNMT enhances renal fibrosis by regulating the DNA methylation of fibrotic genes and acetylation of NF-kB in the mouse model [229]. These data indicate that NAD^+^ metabolism has an important role in CKD progression; however, how NAD^+^ metabolites influence CKD remains unclear. 

### 8.3. Cellular Senescence in Kidney Dysfunction 

Cellular senescence is defined by a permanent growth arrest of damaged cells with commonly enlarged, flattened morphology, which secretes proinflammatory factors through SASP, resistance to apoptosis, senescence-associated heterochromatic foci (SAHF), DNA segments with chromatin alterations reinforcing senescence, the expression of senescence-associated beta-galactosidase (SA-β-gal), the expression of CDK inhibitor p16 and p21, the activation of p53, the expression of high-mobility group A (HMGA) proteins, and reduced lamin B1, among others. A high level of variation has been obtained in the senescent phenotype in relation to the cell type and how the senescent is activated. Cellular senescence has been proven to be an essential cause of many diseases, including cardiovascular, liver, and kidney diseases [230,231]. In human and animal models, CKD shares many phenotypic likenesses with systemic aging. An elevated P16 and SA-β-gal activity level have been observed in several stages of CKD and other kidney diseases, including IgA nephropathy, membranous nephropathy, focal segmental glomerular sclerosis, minimal change disease, and diabetic nephropathy [232]. 

Furthermore, studies have shown that the number of senescent cells is elevated in several departments of the kidney during aging and kidney dysfunction [208,233]. 

Consequently, suppressing cellular senescence may be a likely application to prevent the occurrence of CKD [234]. The abnormality occurs when senescent cells are not discarded by the immune system, have been persistently generated by nearby cells via expression of SASP, or are subjected to chronic injury. Senescent cells cause great damage to neighboring cells via SASP. In addition, a malfunction in the tissue repair may elevate SASP factors, which may quicken the senescence process.

## 9. Cellular Senescence in the Kidney and the Effect of NAD^+^

Cell cycle arrest and SASP through senescence are two main pathways that are significantly influenced by NAD^+^. Narita et al. showed that NAD^+^ stimulates components in SASP through NAMPT, the enzyme of the NAD^+^ salvage pathway in mice [235]. The role of NAD^+^ in proinflammatory SASP through NAMPT is evaluated by biomarkers such as SA-β-gal and P16 [236]. Furthermore, NAMPT and its upstream target HMGA1 were also demonstrated to promote oncogene-induced senescent (OIS) in human fibroblasts [235]. Although, after the stabilization of OIS, the knockdown of HMGA1 or NAMPT showed no reverse effect of senescence [236]. It has also been suggested that NAD^+^ promotes the proinflammatory SASP by affecting glycolysis and mitochondrial respiration. A decrease in the level of NAD^+^/NADH causes a reduction in the glycolysis rate, which was also observed in OIS cells with decreased NAMPT levels [237]. This observation indicates that activation of NAMPT by supplementation with NAD^+^ may be key for proinflammatory SASP in OIS or senescence induced by chemotherapeutic agents.

Thereby, senescent cells are also involved in the NAD^+^ metabolism as SASP causes deprivation of NAD^+^ with age by activation of CD38 in macrophages, a main consumer of NAD^+^ [238,239]. This indicates that senescent cells result in a decline in the level of NAD^+^, which cause more senescence-dependent age-related conditions, e.g., diabetes [13].

An elevated level of NAD^+^ is shown to lower the activation of AMPK and p53 and allow p38MPK to stimulate NF-κB, which causes a transcription of the proinflammatory SASP factors [236]. However, after the establishment of senescent cells, the elevated level of NAD^+^ may alternatively increase the removal of senescent cells through the SASP, as an increase of NAD^+^ can cause activation in PARP and Sirt1. A lack of Sirt1 activity is associated with several aspects of senescence, including SASP activation [240] and cell cycle arrest [241], as well as senescence-associated degenerative pathologies including neurodegeneration, cachexia, fatty liver, and atherosclerosis [5,86]. Therefore, protected NAD^+^ homeostasis and CD38 inhibition obtained by removing senescent cells or suppressing SASP can protect and prevent elevated senescence with age. 

Furthermore, autophagic flux restoration and intracellular NAD^+^ elevation are pathways by which AMPK activation protects cells from oxidative stress-induced senescence [242]. Niacin and resveratrol also reduce cellular senescence formation by activating Sirt1. As Sirt1 is shown to suppress NF-κB signaling, SIRTs also reduce cell cycle arrest by modulating p53, NF-κB, STAT, FOXO-1, and FOXO-3 [243,244]. 

Moreover, exogenous extracellular NAMPT was reported to promote the expression of profibrotic molecules in various types of renal cells [245,246], and an overexpression of endogenous NAMPT presumably induces glomerular inflammation and fibrosis [245]. In addition, NAMPT can directly induce toll-like receptor 4 (TLR4)-mediated NF-κB p65 [247]. This finding indicates that NAMPT may have a crucial role in the physical state of the cell, and an overexpression of NAMPT may promote pathological changes. An abnormal upregulation of endogenous NAMPT may lead to an imbalance between Sirt1 and NF-κB p65, further accumulating cell damage that has been triggered by oxidative stress and eventually leading to cellular dysfunction and apoptosis [245]. In an in vitro study, FK866 or quercetin significantly reduced the expression of NAMPT at the protein level. 

Furthermore, FK866 or quercetin inhibited NMN accumulation in mesangial cells cultured under high glucose conditions. FK866 and quercetin can also enhance the expression of Sirt1 and NMNAT, thereby being able to regulate the NAD^+^ metabolite [248]. This finding suggested that inhibition of NMN accumulation may be a promising target for kidney senescence. 

## 10. NAD^+^ as a Potential Pharmacological Option for AKI and CKD

Genetical approaches in preclinical studies, such as kidney-specific overexpression of PGC1-α and Sirt1 and inhibition of CD38 and PARP by specific inhibitors in animal models, are shown to increase the level of NAD^+^ and improve the cellular homeostasis, which attenuates renal diseases [107,132,144,150]. An increase in the level of NAD^+^ has shown to be beneficial in animal models of human AKI and some CKD models, which have encouraged several clinical trials of NAD^+^ boosters in humans (Table 2). Most of these studies have included precursors such as NAM or NR, which have been mainly studied in relation to AKI and CKD. In addition, NA, a well-known pharmacological drug for treating Pellagra, is also in clinical trials as a potential NAD^+^ booster for treating AKI and CKD (Table 1). NA is also known to lower dyslipidemia and serum phosphorus levels and has adverse effects in patients with CKD [249]. The benefit of NA is that the side effects following NA administration are detailed compared to other precursors [250]. 

Phase I of NR and NAM clinical trials has focused on bioavailability and safety. NRPT, a combined component of NR and pterostilbene, is shown to elevate the whole blood of NAD^+^ levels in mild AKI. In addition, it is proven to be safe and well tolerated in a dose of 1000 mg/200 (NR/ PT) mg twice a day for 48 h in a clinical trial (NCT03176628) for treating mild AKI patients. However, further studies are needed to see the potential therapeutic benefit of NRPT in AKI [42]. Furthermore, a clinical translation study in cardiac surgery patients (NCT02701127) showed that patients who developed AKI had a higher urine quinolinate level [41]. In this study, patients were treated with three doses of either 1 g or 3 g of oral NAM or placebo per day (a day before the surgery, on the day of surgery, and the day after the surgery). Patients treated with 3 g had an elevated level of NAM and NMN in their serum and an increased level of NAM in their urine. Treatment in both groups with 1 g and 3 g of NAM resulted in reduced AKI events. However, clinical trials on CKD by treatment with combined NAM (NCT02258074) as well as a phase IV study with niacin/placebo (NCT00852969) showed no difference in outcomes [28]. NAD boosters such as NR and NAM have shown great pharmacological potential, as they may increase healthy kidneys and prevent AKI; however, the results have been inconsistent in the case of CKD (Table 2). 

## 11. Summary

Overall, the de novo NAD^+^ synthetic pathway is compromised in experimental models and human cells of AKI, while NAD^+^ replenishment attenuates AKI. An elevated NAD^+^ pool can be obtained by supplementation with different NAD^+^ precursors or by inhibiting NAD^+^ consumption pathways. Of note, administration of NAD^+^ boosters have shown conflicting results in preventing CKD progression in patients and animal models. Therefore, further investigations of NAD^+^ in kidney diseases are required, especially focusing on the roles of NAD^+^ in severe AKI, CKD, and senescent cells; senescent cells probably have a significant effect on kidney fibrosis, and their actions appear to be controversial in the case of inhibition or in the case of improvement in NAD^+^ boosting. Additionally, it is essential to investigate and optimize the dosage and time point for NAD^+^ boosting, as NAD^+^ levels and the extent of depletion may vary between cell types. As many of the NAD^+^ precursors such as NR and NMN are bioavailable and clinically safe [164,251], we are cautiously optimistic on the potential clinical applications of NAD^+^ replenishment in treating kidney diseases.

## Figures and Tables

**Figure 1 ijms-24-00137-f001:**
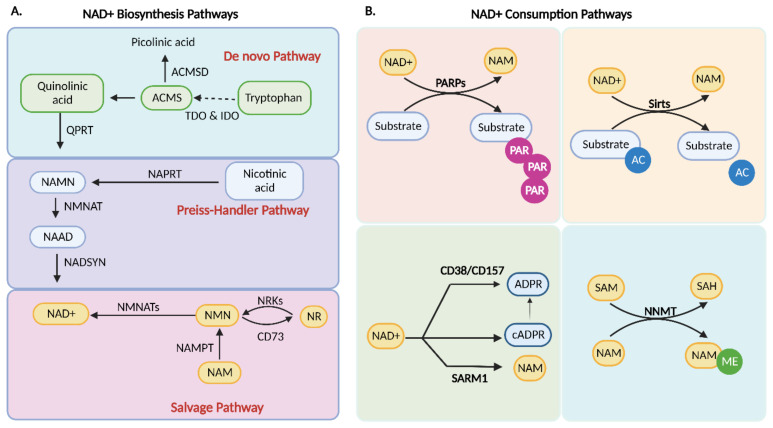
Overview of the NAD^+^ homeostasis. (**A**) The three pathways of NAD^+^ biosynthesis. Mammalian cells synthesize NAD^+^ from tryptophan via the de novo pathway or nicotinic acid via the Preiss–Handler pathway. However, most NAD^+^ is recycled by salvage pathways from nicotinamide (NAM), a byproduct of NAD+-consuming reactions. (**B**) The four known NAD^+^ consumption pathways. NAD^+^ acts as a co-substrate for many enzymes, including PARPs, SIRTs, CD38/CD157, and SARM1, affecting metabolism, genomic stability, gene expression, inflammation, circadian rhythm, and stress resistance. Utilizing NAD^+^ as a co-substrate for PARPs and SIRTs regulates their target molecules, producing NAM as a byproduct. The CD38/CD157 and SARM1 also utilize NAD^+^ and generate NAM, ADPR, and cADPR. Furthermore, the enzyme NNMT catalyzes the transfer of a methyl group from SAM to NAM, which result in S-adenosylhomocysteine (SAH) and methyl nicotinamide. IDO: indoleamine 2,3-dioxygenase; TDO: tryptophan DO; ACMS: α-amino-β-carboxymuconate-ε-semialdehyde; ACMSD: ACMS decarboxylase; QPRT: quinolinate phosphoribosyltransferase; NAPRT: nicotinic acid PRT; NAMN: nicotinate mononucleotide adenylyl transferases; NAAD: nicotinic acid adenine dinucleotide; NADSYN: NAD synthase; NR: nicotinamide riboside; NRKs: NR kinase 1/2; PARPs: poly (ADP-ribose) polymerases; NNT: nicotinamide nucleotide transhydrogenase; SARM1: sterile alpha and TIR motif containing 1; NNMT: nicotinamide N-methyltransferase; NMN: nicotinamide mononucleotide; NAM: nicotinamide; SAM: S-adenosylmethionine; SAH: S-adenosylhomocysteine; MNA: methyl nicotinamide.

**Figure 2 ijms-24-00137-f002:**
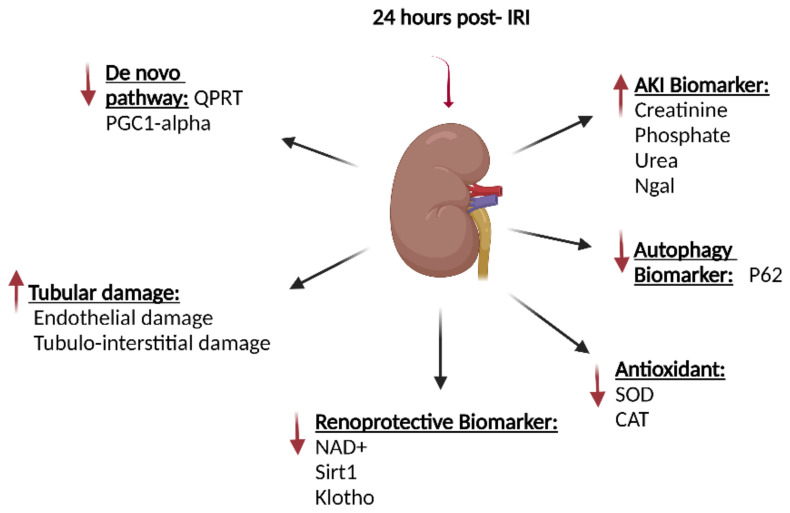
Alternated NAD^+^ related pathways in IRI-induced AKI rats after 24 h. The IRI-induced AKI resulted in a significantly elevated creatinine, phosphate, and urea and an increase in the *Ngal*, all biomarkers for AKI. The tubular damage is also observed at 24-h post-IRI. This model showed a significant decrease in the biomarkers for renal health, NAD^+^, *Sirt1*, Klotho, and antioxidants such as *Sod2* and *Cat*. *Qprt* and *Pgc1-α* were also downregulated in this model, the two genes involved in the de novo biosynthesis pathway. An elevated level of P62 is also observed, a biomarker for autophagy. IRI: ischemia-reperfusion injury; AKI: acute kidney injury; *Ngal*: neutrophil gelatinase-associated lipocalin; *Sirt1*: Sirtuin1; *Sod2*: superoxide dismutase; *Cat*: *catalase*; *Qprt*: quinolinate phosphoribosyltransferase; *Pgc1-α*: peroxisome proliferator-activated receptor-gamma coactivator-1 alpha.

**Figure 3 ijms-24-00137-f003:**
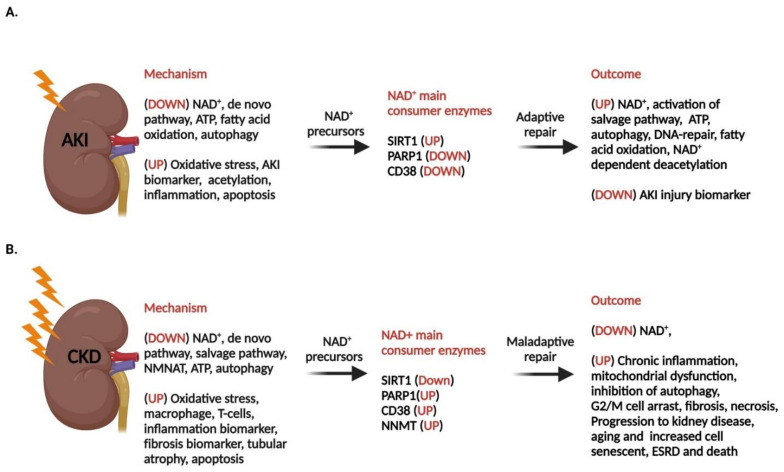
Compromised NAD^+^ in response to AKI and CKD and augmentation with NAD^+^ precursor. (**A**) Compromised NAD^+^ in mild ischemic-induced AKI leads to an impaired NAD^+^ de novo biosynthesis pathway. However, augmentation with NAD+ improves Sirt1 deacetylase activity and activates PARP1 for DNA repair, resulting in the kidney’s adaptive repair. (**B**) CKD or severe AKI are shown to impair de novo and perhaps also salvage pathways. Therefore, augmentation with NAD+ does not significantly increase the level of NAD^+^. Furthermore, the high level of tissue damage will further lead to a reduced level of Sirt1 and DNA repair capacity by dysregulating PARP1. NAD^+^ consumption will further increase by an elevated level of CD38, which can increase the level of NNMT and promotes acetylation and activation of several transcription factors, leading to aggravated effects such as inflammation, apoptosis, mitochondrial dysfunction, fibrosis, oxidative stress, and autophagy dysfunction in the kidney. Together, these processes contribute to the maladaptive repair of the outcome and the development and progression of kidney-related disorders. AKI: acute kidney injury; CKD: chronic kidney disease; Sirt1: Sirtuin 1; PARP1: poly (ADP-ribose) polymerases, and NNMT: nicotinamide N-methyltransferase.

**Table 1 ijms-24-00137-t001:** NAD^+^-based therapies and their effects on some AKI and CKD studies.

Disease/Condition	NAD^+^ Boosters Compound	Outcome Observed after Genetic Modulation/NAD^+^ Boosting	Ref.
AKI			
Model 1: IRI-induced mice: - Real NAD^+^, NADH, and QPRT (**DOWN**) - Renal and urinary Quin and Q:T (**UP**) Model 2: Renal *Qprt-/+* mice: - Renal NAD^+^ and de novo pathway (**DOWN**) - sCr, AKI susceptibility, tubular necrosis, and renal and urinary Quin and Q:T (**UP**)Model 3: Ischemic human kidney: - Urinary Quin and Q:T (**UP**) Model 4: Cardiac surgery patients: - AKI susceptibility, sCr, and Troponin T (**UP**)	Genetically approaches by down-regulation QPRT or pharmacological approach by NAM	Model 1: Wild-type control mice: - Real NAD^+^, NADH, and QPRT (**UP**) - Renal and urinary Quin and Q:T (**DOWN**) Model 2: Wild-type control mice: - Renal NAD+ and de novo pathway (**UP**) - sCr, AKI susceptibility, tubular necrosis, and renal and urinary Quin and Q:T (**DOWN**) Model 3: Healthy human kidney: - Urinary Quin and Q:T (**DOWN**) Model 4: Cardiac surgery patients: - AKI susceptibility, sCr, and *Troponin T (**DOWN**)	[41]
Knockout of *Pgc1-α* in IRI-induced mice - sCr, DAGs, TAGs, tubular injury in cortex and medulla (**UP**) - Relative renal NAM (**DOWN**)	Genetically approaches by overexpression of *Pgc1-α* alone or in combination with NAM	- sCr, DAGs, TAGs, Tubular injury in cortex and medulla (**DOWN**) - Relative renal NAM, de novo NAD biosynthesis, renal NAD, #β-OHB, and #PGE2 (**UP**)	[107]
Model 1: IRI-induced mice: - pCr, BUN, tubular necrosis, dilation, cast formation, and MPO activity (**UP**) - GSH and NAD^+^ (**DOWN**) Model 2: Cisplatin-induced mice: - pCr, BUN, and renal KIM1 protein, tubular necrosis, dilation, edema, cast formation, and inflammatory cell (**UP**) - GFR and NAD^+^ (**DOWN**)	Pharmacological inhibition of ACMSD (TES-991 and TES-1025)/NAD^+^ Biosynthesis modulators	Model 1: IRI-induced mice: - pCr, BUN, tubular necrosis, dilation, cast formation, and MPO activity (**UP**) - GSH and NAD^+^ (**DOWN**) Model 2: Cisplatin-induced mice: - pCr, BUN, and renal KIM1 protein, tubular necrosis, dilation, edema, cast formation, and inflammatory cell (**UP**) - GFR and NAD^+^ (**DOWN**)	[37]
Model 1: Cisplatin-induced 20-month-old mice and 3-month-old mice: - BUN, sCr, damaged tubules, ac-FOXO-1, C-CAS 3 (**UP**) - Mitochondrial density, NAMPT, NMNAT1, and NMNAT3, NAD^+^, Sirt1 (**DOWN**) Model 2: *Sirt1-/+*, Cisplatin-induced mice:- BUN, sCr, damaged tubules, C-CAS 3 and 9, BAX, (**UP**) - Mitochondrial density, expression of Sirt1 (**DOWN**) Model 3: IRI-induced AKI mice: - BUN, sCr, damaged tubules (**UP**)	NMN	Model 1: 20-month-old and 3-month-old control mice: - BUN, sCr, damaged tubules, ac-FOXO-1, C-CAS 3 (**DOWN**) - Mitochondrial density, NAMPT, NMNAT1, and NMNAT3, NAD^+^, Sirt1 (**UP**) Model 2: *Sirt+/+*, Cisplatin-induced mice: - BUN, sCr, damaged tubules, C-CAS 3 and 9, BAX, (**N.S.)** - Mitochondrial density, expression of Sirt1 (**N.S.**) Model 3: IRI-induced AKI mice: - BUN, sCr, damaged tubules (**DOWN**)	[119]
**CKD**			
UUO-induced renal interstitial fibrosis mice: Collagen, FN, α-SMA, tubular atrophy, C-CAS3, macrophage, T-cells, IL-1 beta, TNF-α (**UP**)	NAM	UUO-induced renal interstitial fibrosis mice: Collagen, FN, α-SMA, tubular atrophy, C-CAS3, macrophage, T-cells, IL-1 beta, TNF-α (**DOWN**)	[27]

AKI: acute kidney injury; QPRT: quinolinate phosphoribosyltransferase; Q:T: quinolinate/tryptophan; Quin: quinolinate; sCr: serum creatinine; NAM: niacinamide; *Troponin: a cardiac injury marker; PGC-1α: peroxisome proliferator-activated receptor gamma coactivator 1-alpha; IRI: ischaemia reperfusion injury; DAGs, TAGs: renal di-/tri-acylglycerols; #(β-OHB: beta-hydroxybutyrate; PGE2: prostaglandin): fat breakdown product; pCr: plasma creatinine; BUN: blood urea nitrogen; MPO: myeloperoxidase; GSH: glutathione; KIM1: kidney injury molecule-1; GFR: glomerular filtration rate; ACMSD: alpha-amino-beta-carboxy-muconate-semialdehyde decarboxylase; ace-FOXO-1: acetylated FOXO-1; NAMPT: nicotinamide phosphoribosyltransferase ; NMNAT1 and NMNAT3: nicotinamide mononucleotide adenylyltransferase1/3 ; Sirt1: Sirtuin1; C-CAS3: cleaved caspase-3; NMN: Nicotinamide mononucleotide; CKD: chronic kidney disease; UUO: unilateral urethral obstruction; FN: fibronectin; α-SMA: α-smooth muscle actin; TNF-α: tumor necrosis factor alpha.

**Table 2 ijms-24-00137-t002:** A partial summary of human clinical trials of NAD^+^ precursors in relation to AKI and CKD (clinicaltrials.gov accessed on 2 December 2022).

NAD^+^ Precursors	Condition/ Disease	Dose Administration	Duration of Treatment	Age/ Sex	Study Title	Phase	Status	References/NCT
NR	AKI	4X Basis™ capsule, BID, PO, (Each capsule: 125 mg of NR and 25 mg of PT)	8 weeks (2 weeks pre-surgery and 6 weeks post-surgery.	18+/ All	Protection from acute kidney injury (AKI) with Basis™ treatment	II	Recruiting	NCT04342975
NR	AKI	2X capsule, BID, PO, (Each capsule: 250 mg of NR)	10 days	18+/ All	Nicotinamide riboside in SARS-CoV-2 (COVID-19) patients for renal protection (NIRVANA)	II	Active, not recruiting	NCT04818216
NR	AKI	2X capsule, BID, PO,(Each capsule: 250 mg of NR and 50 mg of PT)	2 days	18+/ All	Pharmacokinetics, pharmacodynamics and safety of basis in acute kidney injury study (BAKIS)	N/A	Completed	NCT03176628
NAM	AKI	3 g/day NAM, PO	3 days	18+/ All	NAD^+^ augmentation in cardiac surgery associated myocardial injury trial (NACAM)	II	Recruiting	NCT04750616
NAM	AKI	NAM (500 mg mixed in 50 mL of 0.9% saline), every 12 h, intravenously	3 days	18+/ All	Does high-dose vitamin B3 supplementation prevent major adverse kidney events during septic shock? (VITAKI)	III	Recruiting	NCT04589546
NAM	AKI	1 g or 3 g per day of NAM, PO	Baseline and days 1 through 4	18+/ All	Molecular effects of vitamin B3 (Niacinamide) in acute kidney injury	I	Completed	NCT02701127
NA	AKI	1X vitamin B complex, every 12 h, PO	5 days	18–100/All	Intravenous administration of vitamin B complex improves renal recovery in patients with AKI (VIBAKI)	IV	Recruiting	NCT04893733
MIB-626	AKI	1 g MIB-626, BID, PO	14 days	18+/ All	Phase 2a MIB-626 vs. placebo COVID-19	II	Recruiting	NCT05038488
NR	CKD	500 mg NR, BID, PO	3 months	35–80/ All	Nicotinamide riboside supplementation for treating arterial stiffness and elevated systolic blood pressure in patients with moderate to severe CKD	II	Recruiting	NCT04040959
NR	CKD	1 Tablet of 600 mg NR and 2 tablet of 250 mg Coenzyme Q10, BID, PO	6 weeks	30–79/ All	Trial of nicotinamide riboside and Co-enzyme Q10 in chronic kidney disease (CoNR)	II	Completed	NCT03579693
NAM	CKD	1 capsule NAM of 750 mg, BID, PO and 2* 500 mg capsules of lanthanum carbonate with each meal	12 months	18–85/ All	The COMBINE study: the CKD optimal management with BInders and Nicotinamide (COMBINE)	II	Completed	NCT02258074
NA	CKD	1000 mg/day, PO	14 weeks	21–80/ All	Niacin and endothelial function in early CKD	IV	Completed	NCT00852969

BID, twice a day; PO, oral administration; AKI: acute kidney injury; CKD: chronic kidney disease; NR: nicotinamide riboside; NAM: nicotinamide; NA: niacin; MIB-626: NAD-boosting drug; PT: pterostilbene; N/A: not applicable.

## Data Availability

Not applicable.

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
