# Peer review of "Roles of NAD+ in Acute and Chronic Kidney Diseases"

_ijms, 2022, doi:10.3390/ijms24010137_

Round 1

Reviewer 1 Report

The review have eloquently discussed the role of NAD in chronic kidney diseases. I enjoyed reading this review and this should be of interest to people who are studying chronic kidney disease. The authors have given all the necessary references for the review and discussed all the necessary background information regarding this topic.

Author Response

Thank you so much for your time and review. We are happy you found our manuscript helpful for chronic kidney disease studies. 

Reviewer 2 Report

Morevati with colleagues presents a narrative review paper aiming to discuss the interplay between NAD+ biology and both, acute and chronic disease states.

I found the paper well-written and nicely organized as well as discussing an important area in the nephrological and biochemical field. Moreover, the review of the literature is truly comprehensive, thus the paper is more valuable.

I have some majors that should be fixed during the review process:

I would like the introduction of the main figure that presents the role of NAD+ activity/biology in kidney disease pathogenesis at glance - the division into AKI and CKD - that's the main focus of the paper. Readers must be able to differentiate the role of NAD+ in AKI and CKD. It will bring more attractiveness to the readers of the paper. Right now, one image might show more than 1000 words. Moreover, a new table showing the main differences in NAD+ biology in AKI vs CKD should be added.

Moving further, there is a very limited amount of information given regarding the following PRISMA guidelines and description of the searching strategy, which is so crucial for review papers, even narrative reviews. The Author should include at least: Data sources and searches, Study eligibility criteria, Study selection process, Data extraction, and study quality assessment (assessing the risk of bias (ROB) for each included study), Data synthesis. MeSH terms (in addition/replacement of keywords) are necessary to be included. For each step, it is necessary to explain to the reader with pictures or tables. It is necessary to explain what was drawn at each step to lead to the result. Moreover, a figure showing the PRISMA-based workflow must be drawn accordingly to the Prisma schema. After that, a discussion is valuable even for narrative papers. A description of the Data Mining strategy should also be included.

Please describe the possibility of pharmacological interventions in NAD+-related pathways that might be beneficial/crucial for kidney disease patients.

The paragraph on the interplay between main uremic toxins and NAD+ - is an emerging field of nephrology and toxicology nowadays. 

There are some minor grammatical errors / minor typos - please go over them carefully.

Line 365 - a different type of font and size used. Line 586 - the same issue. 

To sum up - I am very supportive of the publication, albeit when the above-listed things are done. 

Author Response

We are glad and appreciate receiving your constructive and valuable comments on our manuscript. We found the comments very thorough and well-noted, which has improved the manuscript by several folds. Furthermore, we deeply believe that the reviewer’s comments and suggestions have increased the scientific value of the revised manuscript. 

Our team worked hard to evaluate and incorporate the comments in the version. The changes came across in the new manuscript in different chapters, figures, tables, and texts. However, we tried to keep the main body and structure of the manuscript as it was in the first version. Along with the new version of the manuscript, our team has replied to each comment, listed in the attached file in italic-red

Please find our replies to your comments enclosed.

Reviewer 3 Report

Interesting paper with well-described mechanisms of nicotinamide adenine dinucleotide in AKI and CKD. I have only some minor remarks.

- The full description is quite redundant and somewhat challenging to read. I suggest to reduce some parts adding the study decription in two tables for NAD+effect on AKI and CKD respectively (i.e. name of the study, supposed mechanisms, outcomes on AKI/CKD etc.) Three main chapters should be considered: NAD+ general description, NAD+ in AKI, and NAD+ in CKD, with subsequent subchapters. 

- The paragraph about klotho is too long and should be incorporated in one of the subchapters considering that an essential part of the description included considerations about senescence.

- Consider and eventually include an appropriate discussion of the following paper (Liu X et al. Impaired Nicotinamide Adenine Dinucleotide Biosynthesis in the Kidney of Chronic Kidney Disease. Front Physiol 2021 doi: 10.3389/fphys.2021.723690)

Author Response

(The authors gave the same response as above.)

Reviewer 4 Report

This is well written, comprehensive review, pointing and discussing several physiologic and pathologic pathways of kidney senescence, post-injury recovery and /or maladaptive repair, fibrosis and development of chronic kidney disease, regardless the underlying cause. 

The only issue, which might of value, especially for clinicians, would a  brief (separate) summary of several therapeutic modalities, which are evaluated in animal setting and/or in humans in terms of preventing/slowing the processes, so attractively described in this manuscript. I suggest to add a table, mentioning several drugs and molecules, tested in clinical trials of phase I to III, which look promising for the future practice. Some relevant  data were mentioned in the text, however a short summary at the end of the manuscript will attract attention of the readers. 

Author Response

(The authors gave the same response as above.)

Round 2

Reviewer 2 Report

The Authors made astonishing work during the peer review process and significantly improved the whole manuscript. They correctly addressed all my comments (both, majors, minors) and the "readability" of the paper will be truly sky-high. 

The last thing to improve is minor lackings in citations of very relevant papers that should be included since they were published recently in top journals - it will bring even more interest to the Readers:

- additional support of interplay between NAD+ and sirtuins -https://doi.org/10.1038/s41374-021-00599-1

- a fresh paper published few days ago that supports Author's conclusions: 

Exp Mol Med (2022). https://doi.org/10.1038/s12276-022-00894-x

- and the paper supporting the role of Sirtuins in CKD pathology:

https://doi.org/10.3389/fphys.2018.01623

Once gain, congrats for your hard and elegant work!

Author Response

Dear Reviewer, 2

Once again, thank you very much. We are glad and appreciate your valuable suggestion and comments on our manuscript. We found the suggested references very relevant and well-noted, which has improved the manuscript more. Furthermore, we thoroughly believe that the reviewer’s comments and suggestions have increased the scientific value of the revised manuscript. 

Our team has evaluated and incorporated the references in the new version. Along with the new version of the manuscript, our team has replied to each comment, listed below in italic

The response to the comments is enclosed.

Sincerely,

Marya Morevati, PhD,                                              

Nephrological Department P 2131,

Rigshospitalet,

9 Blegdamsvej,

2100 Copenhagen,

Denmark.

Phone: +45 35452777

Mobile Phone: +45 50467285

E-mail: marya.morevati@regionh.dk
